# Kitchen Area Air Quality Measurements in Northern Ghana: Evaluating the Performance of a Low-Cost Particulate Sensor within a Household Energy Study

**Evan R. Coffey** [1], **David Pfotenhauer** [1], **Anondo Mukherjee** [2], **Desmond Agao** [3], **Ali Moro** [3], **Maxwell Dalaba** [3], **Taylor Begay** [1], **Natalie Banacos** [4,*], **Abraham Oduro** [3], **Katherine L. Dickinson** [4] and **Michael P. Hannigan** [1]

[1]   College of Engineering and Applied Science, University of Colorado Boulder, 427 UCB, 1111 Engineering Drive, Boulder, CO 80309, USA

[2]   Department of Atmospheric and Oceanic Sciences, University of Colorado Boulder, 311 UCB, Boulder, CO 80309, USA

[3]   Navrongo Health Research Centre, War Memorial Hospital, Upper East Region, Navrongo Box 34, Ghana

[4]   Colorado School of Public Health, University of Colorado Anschutz Medical Campus. 13001 East 17th Place, Aurora, CO 80045, USA

*   Correspondence: natalie.banacos@cuanschutz.edu

**Abstract:** Household air pollution from the combustion of solid fuels is a leading global health and human rights concern, affecting billions every day. Instrumentation to assess potential solutions to this problem faces challenges—especially related to cost. A low-cost ($159) particulate matter tool called the Household Air Pollution Exposure (HAPEx) Nano was evaluated in the field as part of the Prices, Peers, and Perceptions cookstove study in northern Ghana. Measurements of temperature, relative humidity, absolute humidity, and carbon dioxide and carbon monoxide concentrations made at 1-min temporal resolution were integrated with 1-min particulate matter less than 2.5 microns in diameter ($PM_{2.5}$) measurements from the HAPEx, within 62 kitchens, across urban and rural households and four seasons totaling 71 48-h deployments. Gravimetric filter sampling was undertaken to ground-truth and evaluate the low-cost measurements. HAPEx baseline drift and relative humidity corrections were investigated and evaluated using signals from paired HAPEx, finding significant improvements. Resulting particle coefficients and integrated gravimetric $PM_{2.5}$ concentrations were modeled to explore drivers of variability; urban/rural, season, kitchen characteristics, and dust (a major $PM_{2.5}$ mass constituent) were significant predictors. The high correlation ($R^2 = 0.79$) between 48-h mean HAPEx readings and gravimetric $PM_{2.5}$ mass (including other covariates) indicates that the HAPEx can be a useful tool in household energy studies.

**Keywords:** particulate matter; low-cost sensors; cooking; particle coefficients; household pollution; gravimetric filter

## 1. Introduction and Background

Pollution from the inefficient combustion of biomass is a global environmental health concern. The global burden of disease linked more deaths and disability adjusted life years (DALYs) attributable to household air pollution (HAP) from solid fuels than from unimproved water and sanitation [1]. HAP from solid fuels is the number one risk of morbidity in south Asia, and the third highest risk globally. Particulate matter (PM) emissions are a principal component of solid biomass combustion and exposure to PM less than 2.5 microns in diameter ($PM_{2.5}$) is linked to adverse health outcomes including increased risk of acute lower respiratory infections, chronic obstructive pulmonary disease,

stroke, and cardiovascular and circulatory diseases [2–4]. Interventions tried addressing this issue by replacing traditional, unvented biomass stoves used for cooking and home heating with varying degrees of success. Air pollution monitoring and exposure assessments of proposed solutions to this challenge confront many obstacles, with suitable, cost-effective instrumentation being one [5,6]. Typical solid fuel HAP research settings invoke an array of sampling constraints including cost, power consumption, physical form factor (i.e., size and weight), accuracy, precision, durability, dynamic sensing range, and user-compatibility. Low-cost sensors are becoming ubiquitous in efforts to increase spatiotemporal information in HAP exposure assessments [7]. Particulate sensors are among the most pursued due to the well-studied links between that constituent and adverse health outcomes. However, many PM monitoring instruments cost between 500 and 10,000 United States dollars (USD) and require substantial technical skills and laboratory support, limiting the scope or consideration of air quality assessments, an integral component of evaluating HAP interventions [8]. Time-resolved measurements of PM yield a wealth of information that cumulative measurements lack, and they can play a vital role in discerning exposure from a multitude of sources beyond cooking and heating. Research groups assessing the impacts of HAP-related interventions can more precisely link pollution events tied to specific activities to intervention treatments through the integration of contemporaneously measured data on short time scales [9]. Understanding the quality of these measurements is important.

PM monitors based on the measurement of light scattering can address several HAP research-related constraints. These sensors were extensively evaluated in the laboratory, but less so in the field [5,7,10–12]. Briefly, these monitors operate by emitting light that interacts with particles in a sensing region and by measuring changes in incident light on a photodetector. Relationships are commonly developed to characterize these photoelectric changes with mass concentrations among other varying particle properties (i.e., reflectivity, shape, number, size, density, and chemistry). These sensors are typically linearly calibrated for a given PM mixture with a sensitivity (response per $\mu g \cdot m^{-3}$) and an offset (response in clean air). Because these sensors indirectly measure mass, they require collocations with a pump-and-filter gravimetric standard with which to ground time-resolved readings. Resource (e.g., cost and time) savings can be realized when a subset of total measurements includes low-cost and gravimetric samples. For example, The Gold Standard, a certification entity for climate and development initiatives, requires a correlation of at least 0.75 between 24-h gravimetric and optical monitor measurements for exposure assessments acknowledging variation among site-specific factors (e.g., fuel, stove type, cooking practices, household characteristics, season, etc.) [13]. Standards like these simultaneously hold low-cost tools to certain benchmarks yet allow research groups to tailor instrumentation to their research needs/limitations.

The Prices, Peers, and Perceptions (P3) study was conducted in the Kassena-Nankana District (KND) of northern Ghana between late 2016 and early 2019. The purpose of the study was to examine households' demand for and the impacts of improved cookstoves. Specifically, we conducted two sets of stove price experiments in the rural and urban areas of the KND. The P3 Bio study enrolled 293 rural households and offered them two types of improved biomass cookstoves (the Greenway Jumbo and the ACE1 stove) at prices that were randomized at the group (cluster) level [14].

Meanwhile, the P3 Gas study enrolled a second set of 262 urban households, and offered each participant a set of liquefied petroleum gas (LPG) stove-fuel packages. In both arms, a key aim of the study was to examine variation in household air pollution, particularly $PM_{2.5}$, across groups over time. Thus, developing methods to measure $PM_{2.5}$ concentrations accurately in household cooking areas and for personal exposure was a central focus of our research activities. This paper describes these methods, focusing on comparisons of the performance of low-cost optical sensors with gravimetric benchmarks.

One of our main research objectives utilizing the low-cost monitors was to characterize elevations in $PM_{2.5}$ concentrations above background levels to gain a better understanding of the sources of such deviations. Ambient concentrations of $PM_{2.5}$ in our study region, albeit relatively moderate as measured by others in this region [15], have potential significant overall health implications

and would not be well captured by the low-cost monitors. Rather, our use of the low-cost monitors draws the much-needed attention to understanding the magnitudes and sources contributing most to increases in $PM_{2.5}$ concentrations known to have detrimental health effects, those from nearby sources.

We report a total of 62 48-h cumulative $PM_{2.5}$ gravimetric mass concentrations and constituents (elemental carbon [EC], organic carbon [OC], etc.) from 60 urban and rural kitchens in northern Ghana. This work makes four main contributions to the literature on existing low-cost PM monitoring. Firstly, we introduce and evaluate a simple, yet promising post-collection baseline drift correction algorithm and two signal filters to automate quality control of low-cost monitor data. Secondly, we generate a relative humidity correction described by data provided by Wang et al. [16] for an optical PM sensor and apply it, as well as other corrections found in the literature. Thirdly, we present comparisons of mean low-cost readings to gravimetrically measured $PM_{2.5}$ mass concentrations. Fourthly, we model particle coefficient and total $PM_{2.5}$ mass concentrations with co-measured environmental variables (e.g., CO, $CO_2$, temperature, humidity, EC, OC, etc.), season, urban/rural kitchen classifications, and kitchen characteristics to explain additional variation and explore differences among predictors. Total $PM_{2.5}$ mass concentration is modeled without filter-based information to simulate performance without gravimetric standards.

## 2. Materials and Methodology

### 2.1. Low-Cost PM Tool Selection

Several PM monitors were considered for use in the P3 study. We ultimately chose the Household Air Pollution Exposure (HAPEx) Nano (v.3) monitor based on lab and field performance of the housed optical sensor [16–19] and the integrated sensor platform [20–23], as well as cost, size, and low power consumption.

The HAPEx Nano PM monitor (v3. 159 USD, Climate Solutions Consulting, USA) incorporates the Sharp GP2Y1010 optical sensor (<20 USD), an accelerometer, a microprocessor, and a lithium battery in a small physical form factor (90 × 28 × 50 mm) to passively monitor PM concentrations at user-selected time intervals for up to three years [22]. It was the most affordable tool ($159) at the time we began our study and since decreased in price ($119) while showing promising results in the field [20,21,23] and performing well in the lab [23].

Briefly, the HAPEx Nano's optical sensor (Sharp GP2Y1010) involves a pulsed infrared light-emitting diode (IR-LED, 860–950 nm) and a phototransistor positioned at an angle creating a 60-degree forward-scattering measurement of illuminated particles within the sensing pathway. The HAPEx Nano has no forced air flow or inlet selection mechanism, relying rather upon diffusion and/or natural convection of suspended particles through the sensing pathway. The settings are user-defined via a personal computer (PC) software interface. They were reported to be most sensitive to PM with aerodynamic diameter between 1 and 3 μm [20].

NexLeaf Analytics was funded by the World Bank to perform laboratory and field evaluations of several low-cost PM monitoring tools, including the HAPEx Nano and Particle and Temperature Sensor Plus (PATS+) (Berkeley Air Monitoring Group, USA), in an effort to identify and assess the performance of commercially available off-the-shelf PM sensing platforms for large-scale implementation [23]. The lab tests ($n_{HAPEx}$ = 8) involved collocating low-cost monitors inside a custom mixing chamber alongside a GRIMM optical particle counter (OPC) and a gravimetric pump and filter sampler, while exposed to wood-burning emissions at 12 distinct $PM_{2.5}$ mass concentration levels (426–6232 μg·m$^{-3}$). The resulting linear regression coefficient of determination ($R^2$) between HAPEx and gravimetric measurements was 0.80 with an overall particle correction coefficient of 5.68 ($PM_{2.5}$ μg·m$^{-3}$/HAPEx reading) and intercept of 69 μg·m$^{-3}$. Linearity between the HAPEx monitor and the OPC was quite high ($R^2$ = 0.99). The field evaluation involved collocating the low-cost sensor modules with gravimetric pump and filters within biomass-burning kitchens in the Odisha region of India over one season ($n_{HAPEx}$ = 18). Three sampling time comparisons were employed (18–22 hourly,

7–8 hourly, and 2–4 hourly) to capture a wider range of cumulative average $PM_{2.5}$ concentrations (95–6124 $\mu g \cdot m^{-3}$). Resulting regressions showed high linearity ($R^2 = 0.75$) eligible for Gold Standard certification and an overall particle correction coefficient of 0.22 (HAPEx reading/$PM_{2.5}$ $\mu g \cdot m^{-3}$).

The Sharp GP2Y1010, 1–3 orders of magnitudes lower in cost than benchmarked nephelometers and photometers [24], was well described and optical properties rigorously characterized in a laboratory setting [17]. Wang and co-workers found high linearity ($R^2 \geq 0.98$, 0–1 $mg \cdot m^{-3}$) against SidePak-measured $PM_{2.5}$ concentrations and high precision among other Sharp sensors without convective flow [16]. They also characterized lower detection limits, as well as particle size, composition, relative humidity, and temperature effects on the raw signal output, finding high dependence on several factors.

Budde and colleagues performed lab testing of the Sharp GP2Y1010 alongside the TSI DustTrak DRX Aerosol Monitor 8533 (~9000 USD) in a non-controlled indoor environment, finding mean absolute errors (MAE) of 20 $\mu g \cdot m^{-3}$ using 5-min paired readings over a range of 0–150 $\mu g \cdot m^{-3}$ [24]. They also found that time-averaging above 5 min did not drastically improve agreement between the Sharp and the TSI instrument. One-minute comparisons did have relatively larger MAE (20–50 $\mu g \cdot m^{-3}$).

Berkeley Air Monitoring Group commercially packaged Sharp's more precise sister sensor, the GP2Y1014, into the PATS+ (~500 USD), which operates on the same optical measurement principles but which utilizes custom firmware advances to more precisely control current that drives the IR-LED at two distinct intensities, giving way to two separate measurement sensitivity domains and, thus, measurement ranges [7]. Laboratory testing of the PATS+ collocated with a gravimetrically adjusted TSI DustTrak II Aerosol Monitor 8530 inside a controlled chamber resulted in high linear agreement ($R^2 \geq 0.99$) at both sensitivity domains (0–5 and 0–40 $mg \cdot m^{-3}$) for 15-min mean values. When deployed alongside gravimetric samplers in wood-burning Guatemalan kitchens for 48 h, the PATS+ mean $PM_{2.5}$ concentration estimates showed strong correlation with integrated gravimetric $PM_{2.5}$ ($R^2 = 0.9$) over a range of about 650 $\mu g \cdot m^{-3}$, showing promise of sensing capabilities in estimating field-based multi-day kitchen area concentrations at the low–mid-cost range [7].

## 2.2. Household Measurements

Microenvironment sampling took place in a subset of P3 Bio and Gas households enrolled in exposure measurements [14], between September 2017 and March 2019. The ongoing sampling generally took place within two 48-h deployment periods weekly; the first deployment period was Monday to Wednesday and the second was Wednesday morning to Friday morning. The sampling apparatus consisted of three main parts: a custom low-cost environmental monitor called the G-Pod, a set of gravimetric pump and filter trains, and a pair of HAPEx loggers. The apparatus was positioned at a height of 1.5 m, 1 m from the most used cooking location according to the primary cook during that time of year (Figure 1).

### 2.2.1. The G-Pod (Continuous Gas Phase Measurements)

The G-Pod is a modified U-Pod sensor platform designed and built by the University of Colorado, Boulder's Hannigan Air Quality and Technology Research Lab (mobilesensingtechnology.com; https://www.colorado.edu/lab/hannigan), used extensively in air quality research applications [9,14,25–41]. The G-Pod represents a version of a fleet of second-generation U-Pods with the capacity to measure carbon monoxide (electrochemical CO-B4; Alphasense, UK), nitrogen oxides (electrochemical $NO_2$-B4 and NO-B4; Alphasense, UK), ozone (metal oxide MICS-2611, SGX, Switzerland, formerly MicroChemical Systems SA), carbon dioxide (nondispersive infrared S-300; ELT, Korea), volatile organic compounds (metal oxide 2600, 2601; Figaro, USA), temperature and humidity (capacitive and band gap SHT25; Sensirion AG, Switzerland), pressure (piezoresistive BMP-180; Bosch Sensortec, Germany), and location (global positioning system (GPS), 63530 Copernicus II; Trimble, USA) for less than 1500 USD. This work focuses on measurements of CO, $CO_2$, temperature, and relative humidity from the G-Pod.

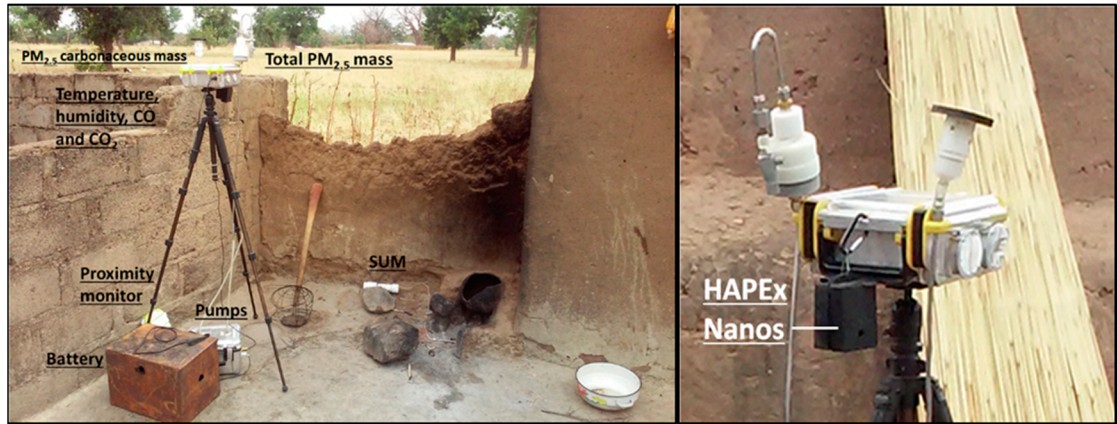

**Figure 1.** Microenvironment sampling apparatus deployed at a typical rural kitchen in the Kassena-Nankana (KN) district. Additional monitoring instrumentation is shown including a stove use monitor (SUM) and participant proximity monitor.

### 2.2.2. Gravimetric Filter Sampling (Cumulative PM Measurements)

Particulate matter pump and filter sampling was performed using two separate sampling trains: one designed to collect $PM_{2.5}$ for total integrated mass and a second to collect $PM_{2.5}$ for carbonaceous $PM_{2.5}$ analysis. Total $PM_{2.5}$ was collected on a 47-mm PTFE (polytetrafluoroethylene) filter (Teflo R2PJ047; Pall, USA) housed downstream of a PTFE-coated aluminum cyclone inlet (2000-30EQ, three liters per minute (lpm) $PM_{2.5}$ cut; URG, USA) connected to a pressure-compensating vacuum pump (Universal PCXR8, SKC Inc, USA) set at a flow rate of $3.0 \pm 0.02$ lpm. A second filter train collected $PM_{2.5}$ mass on a 25-mm, pre-cleaned quartz fiber filter (Pallflex Tissuquartz 2500QAT-UP; Pall, USA) located downstream of an impactor (URG-2000-30PASS-1; URG, USA) positioned inside a filter holder (URG-2000-25F-2LPM 2 lpm $PM_{2.5}$ cut; URG, USA) and connected to a pressure-compensating vacuum pump (Airlite; SKC Inc, USA) set at a flow rate of $2.0 \pm 0.02$ lpm.

### 2.2.3. Low-Cost PM Monitor: HAPEx Nano (Continuous PM Measurements)

Two HAPEx loggers were suspended in open air from the G-Pod roughly 10–20 cm from the pump and filter inlets for the duration of the deployment period. In an effort to capture real-world performance of the HAPEx and to better approximate conditions of the HAPEx worn for personal exposure sampling throughout the study, no enclosures or screens were used to isolate the HAPEx from the environment. The two HAPEx monitors collocated with the G-Pod rotated monthly, providing a wider range of paired readings, capturing differences among a variety of individual HAPEx monitors. HAPEx Nano monitors came factory calibrated for mass concentrations of incense smoke by Climate Solutions Consulting, and calibrations are recommended annually to check monitor sensitivity [22].

### *2.3. Quality Assurance and Quality Control*

### 2.3.1. Gravimetric Filter Sampling and Analysis

Flow calibrations were performed weekly on all sampling pumps using a rotameter calibrated to local conditions using a DryCal flow calibrator (DC-Lite 1.08, Mesa Labs, USA). Any pumps deviating from the setpoint ±0.02 lpm were recorded and corrected. If the pump flow could not be corrected, it was removed from sampling circulation. The PCXR8 pump includes a total sampling time readout which was compared to durations calculated from the enumerator recorded start and stop times.

Quartz fiber filters were conditioned at 500 °C for 18–24 h to remove possible contamination prior to deployment. They were stacked and stored in pre-baked sterile amber glass jars separated by sterile aluminum foil discs. Following sampling, these filters were stored in designated amber glass jars with sterile aluminum foil discs to separate filters. Transportation and storage procedures are described

elsewhere [38]. EC and OC sample concentrations were measured in Boulder, Colorado following the NIOSH 5040 protocol. A 1.5-cm$^2$ filter punch from each sample was analyzed with a Sunset Laboratory OC–EC analyzer using the thermal optical transmittance method (TOT) where total carbon (TC) is the sum of EC and OC. The remainder of the filters were preserved for future organic speciation analysis. Field blank filters, filters that underwent the same handling protocols as 48-h samples yet did not sample any air volume, were placed in each jar, and median concentration values from the set of field blank filters were subtracted from each sample concentration to correct for transportation and handling contamination.

Total PM$_{2.5}$ mass depositions were determined by weighing the PTFE filters before and after sampling using a five-digit LabServe model microbalance (BP210D; Sartorius Corporation, Goettingen, Germany) housed in a custom-built, temperature- and humidity-controlled chamber. The filters underwent a 24-h equilibration process inside the chamber prior to weighing (in triplicate), and the median blank deposited filter mass was subtracted from the deposited sample mass as documented by Dutton and colleagues [42]. Filter mass depositions less than 21 μg (1.5 × SD of blank filter weighing) were flagged as below the detection limit (BDL) and replaced with 140 μg (10 × SD, ~16 μg·m$^{-3}$). Total integrated mass concentrations (TM, μg·m$^{-3}$) and associated propagated error were determined using blank-corrected PTFE mass loadings, sample flow rate, and time. Total organic mass concentration (OM, μg·m$^{-3}$) was estimated from OC using an estimate of the OM/OC ratio of 1.6 found for primary biomass emissions from a mix of traditional and improved cookstoves in the lab [43]. OM/OC ratios typically increase as organic aerosols age and oxidize, yet we assume primarily the collection of fresh emissions. Non-carbonaceous organic mass (NCOM, μg·m$^{-3}$, predominantly oxygen and hydrogen) was estimated as (NCOM = OM − TC), whereas inorganic, non-carbonaceous mass (NCIM, μg·m$^{-3}$) was estimated as (NCIM = TM − (EC + OM)).

### 2.3.2. G-Pod

Gas phase measurements of CO and CO$_2$ concentrations were determined using corresponding sensor signals and regression-based calibration coefficients computed from in-field zero and span gas normalizations, as well as lab-based calibrations using NIST traceable gas standards. More information on the sensor calibration process is documented elsewhere [37,38]. Minute medians of CO, CO$_2$, temperature, and relative and absolute humidity were integrated with time-matched HAPEx readings. Modified combustion efficiency (MCE = $\Delta CO_2/(\Delta CO_2 + \Delta CO)$), a proxy for combustion efficiency, was calculated using elevated CO$_2$ and CO molar concentrations above background. Background CO$_2$ and CO concentrations were determined for each 48-h deployment as the third percentile and minimum, respectively.

### 2.3.3. HAPEx Signal Processing

Indirect measurements of particle mass at high temporal resolution (sub-minute) in dynamic environments require significant signal processing. Li and co-workers explored on-line and off-line GP2Y1010 signal processing techniques, finding a sliding time window and low-pass filter to isolate signal from noise to increase data quality for indoor networks of monitors [19], and their results helped frame our signal processing.

#### Baseline Drift Correction Algorithm and Zeroing

Light-scattering-based particle instruments, especially low-cost models, often require frequent zeroing, or characterization of signal in clean air. Cleaning and zeroing of all the HAPEx monitors took place weekly as monitors were commonly deployed for a week at a time. Briefly, the HAPEx sensing chambers were cleaned by blowing air gently through the inlet in an effort to dislodge adsorbed particles and preserve sensitivity over time. Zeroing was completed by placing all the monitors in a sealed case for 10 min, where air inside the case was first pulled through a filter before entering the case. Our group and many others zero low-cost monitors before and after deployments to the field to capture beginning

and end signal states. The main concern with pre- and post-baseline corrections is that this approach often assumes the drift occurs linearly through time, an assumption we discovered is not accurate. Rather, we identified abrupt (1–5 min) step changes in baseline (background concentration) sensor readings typically coinciding with large spikes in signal within a majority of the 48-h deployments (Figure 2).

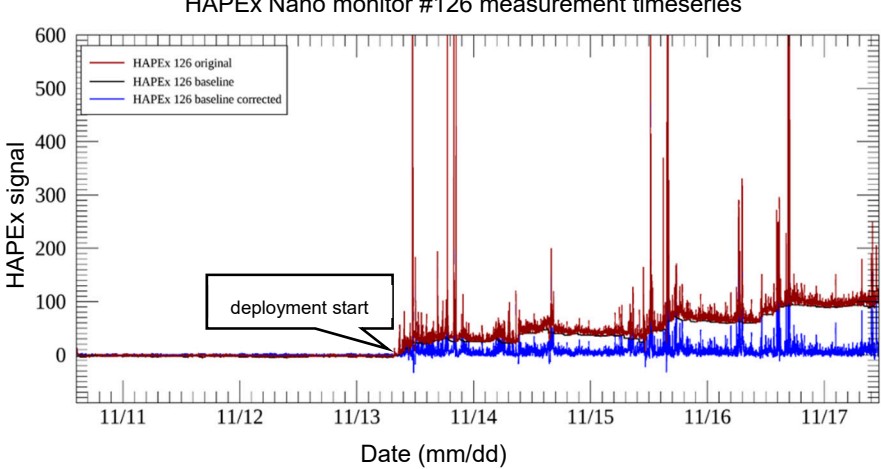

**Figure 2.** Raw (red) and baseline-corrected (blue) Household Air Pollution Exposure (HAPEx) readings using a dynamic baseline (black) calculated from a rolling 80-min window. This monitor was kept in a clean environment from 11 November to the morning of 13 November, and then transported and deployed to study households for two consecutive 48-h deployments.

These step changes can be positive and/or negative in a single deployment. Lens fouling and/or overall sensing chamber reflectivity changes caused by particle adsorption to or desorption from the sensing chamber walls were posited as sources of changes in baseline signals. To address these fluctuations in baseline signal and their associated errors, a baseline drift correction algorithm was created. The objectives of this algorithm were twofold: (1) to distinguish real shifts in signal from false shifts while minimizing the introduction of new errors, and (2) to preserve expected signal distributions. To evaluate the performance of the algorithm, raw and baseline-corrected data logged from paired monitors were assessed for correlation and overall agreement. Briefly, the algorithm sets a baseline value as the third lowest minute value in an 80-min running window (ignoring up to four negative values) and subtracts this value from all raw readings within that window. The mode of the corrected data is set to zero to center the noise for the entire deployment. Histograms of the raw and corrected data were plotted and examined for log-normal distribution. Raw data were preserved alongside corrected data for comparison throughout subsequent analysis.

Base Filter

All paired baseline corrected HAPEx data were filtered for erroneous readings by comparing readings from the two collocated HAPEx. A 5-min rolling average was calculated for each HAPEx. Absolute differences between the two HAPEx rolling readings deviating by more than three standard deviations from the mean difference were flagged as outliers and omitted from the analysis. A cumulative record of the number of outliers was calculated. Because there were no paired readings when only one HAPEx monitor collected data, these readings bypassed this filter.

Threshold Signal Filter

A signal threshold filter was implemented (as an option) for two reasons. Firstly, noise in the HAPEx signal over the course of the sampling campaign was found to be monitor- and deployment-specific, most likely an effect of long-term sensor fouling and variability in environmental variables. Secondly,

meaningful logarithmic transformations of HAPEx data require positive values. The method by which non-positive readings and signals indistinguishable from noise were converted to meaningful values was implemented through this filter. Specifically, this filter involved calculating the fifth percentile of the 5-min rolling standard deviation of each monitor deployment to approximate the noise associated with each unit during the cleanest portion. This value was multiplied by three to approximate signal threshold (ST, raw) above background. The same task was conducted in a known, clean environment during the zeroing procedure where the average ST was 5.4 signal counts. The clean air ST was used in the case a monitor had an ST less than 5.4 during a deployment.

Correlation and agreement between baseline-corrected paired readings and those of the raw readings were evaluated using scatter and Bland–Altman plots, respectively. Lin's concordance correlation coefficient (CCC), which assesses agreement between variables, and the coefficient of reproducibility (RPC) were used as evaluative measures of the baseline correction algorithm.

Relative Humidity Correction

Humidity effects on Sharp GP2Y1010 sensors were examined by others in a lab setting [16]. Holding particle properties and temperature constant, Wang and colleagues subjected several GP2Y1010 sensors, a SidePak Personal Aerosol Monitor AM510 (TSI Inc.), and scanning mobility particle sizer (SMPS, TSI Inc.) to varying levels of relative humidity (RH; 20%, 67%, 75%, and 90%), investigating changes in sensor sensitivity to reference-calculated $PM_{2.5}$ concentrations (0–5 mg·m$^{-3}$). Results from their testing were provided and used to fit a model relating changes in sensitivity (mV per µg·m$^{-3}$) to relative humidity against the SidePak and SMPS reference mass concentrations, independently (Figure S1, Supplementary Materials). Both models show nonlinear relationships; however, when compared against the SidePak, the trend is superlinear, while a quadratic relationship best fits the data when the SMPS is used as the reference. Wang and colleagues noted that the SMPS may not be applicable under high values of RH, since the sheath flow may dry the particles and cause an underestimate of particle concentrations.

Other groups [44–48] empirically determined relative humidity correction factors for higher-end nephelometers, with some finding superlinear relationships similar to results from Wang et al. for the Sharp GP2Y1010. Soneja and coworkers found log-transformed relationships between gravimetric mass and nephelometer readings as a function of RH to have lower root-mean-squared error (RMSE) than non-transformed variables but cautioned against the presence of overfitting [45]. They also recommended correcting over the full range of relative humidity, not just above 60% as originally proposed by Chakrabarti et al. [47].

Two pointwise RH corrections were tested on baseline-corrected 1-min HAPEx data (before base and signal threshold filters applied) over the full RH range to investigate effects on relationships between mean HAPEx and gravimetric $PM_{2.5}$ mass concentrations above and beyond 48-h mean RH. The first, originally discovered by Laulainen [48] and used by Chakrabarti et al., uses Equation S1 and the second, the empirical relationship derived from data provided by Wang and colleagues is expressed in Equation S2. Details on these corrections can be found in the Supplementary Materials.

2.3.4. HAPEx and Environmental Measures of Central Tendency

Kitchen environments experience large variability in environmental factors like temperature, humidity, and CO and $CO_2$ concentrations over 48-h periods and seasonally. Encompassing this variability toward reaching a single comprehensive metric with which to use in regression modeling for each deployment is a challenge and deserves a closer look. Two measures of HAPEx signal central tendency were calculated with which to compare to cumulative filter measurements in terms of arithmetic and geometric means. The geometric mean is less sensitive to large outliers, while the arithmetic mean treats each measurement equally.

Likewise, 48-h CO, $CO_2$, temperature, absolute and relative humidity, and MCE values were calculated as arithmetic averages and as HAPEx-signal weighted-averages reflecting environmental

conditions proportional to HAPEx signal (Equation S3, Supplementary Materials). This is the first time, to our knowledge, this weighted-averaging technique was applied to low-cost PM sensor field data to arrive at environmentally significant parameters for multi-day periods.

*2.4. Modeling Gravimetric PM$_{2.5}$ Concentrations and Particle Coefficients with Low-Cost Sensors*

Optical (i.e., light-scattering) sensors indirectly measure PM mass and, therefore, require gravimetric measurements with which to ground sensor readings. Traditionally, particle coefficients (PC) also known as correction factors, are scaling values used to ground sensor readings to gravimetric measurements (Equation (1)).

$$PC = \frac{mean(HAPEx)}{[Gravimetric\ PM_{2.5}]}. \tag{1}$$

To assess 48-h mean relative humidity effects on PC, Equations (2) and (3) were employed using mean RH and HAPEx-weighted mean RH, respectively. The concept is that a 48-h mean RH term proportional to HAPEx response could explain more variation in PC. Additionally, a log–log-transformed version was employed (Equation (4)) as recommended by Soneja et al. Lastly, a nonlinear model derived by Laulainen and co-workers experimentally [48] and characterized by others [47] was fit using Equation (5). These relationships represent most model forms relating RH to PC found in the literature, but the list is not exhaustive.

$$PC \sim \beta_0 + \beta_1(meanRH) + \epsilon. \tag{2}$$

$$PC \sim \beta_0 + \beta_1\left(HAPEx_{weightedRH}\right) + \epsilon. \tag{3}$$

$$\log(PC) \sim \beta_0 + \beta_1\left(\log\left(HAPEx_{weightedRH}\right)\right) + \epsilon. \tag{4}$$

$$PC \sim a + \frac{b \times \left(HAPEx_{weightedRH}^2\right)}{1 - HAPEx_{weightedRH}} + \epsilon. \tag{5}$$

Moving beyond effects attributable to RH only, a third multilinear regression model estimating PC was informed by stepwise linear regression incorporating interactions among 48-h mean G-Pod measurements (e.g., temperature, and CO and $CO_2$ concentrations) and HAPEx baseline-corrected signal, filter mass constituents (EC, OC, dust etc.), season, urban/rural classification, and kitchen descriptions, which was ultimately reduced to the final model (Equation (6)). "Season" classification was consistent with current and past studies in this region [9,14,15,30,34,35,37,38,40,41,49,50] and "CoverageClass" is a variable used to classify kitchen descriptions to represent a wide range of kitchen area geometries and coverage types (e.g., roof with two walls, no roof with one wall). A breakdown of the "CoverageClass" by urban/rural samples is available in Figure S2 (Supplementary Materials).

$$PC \sim \beta_0 + \beta_1(meanHAPEx \times season) + \beta_2(dust \times season) + \beta_3(meanHAPEx \times LocationType) + \epsilon. \tag{6}$$

Gravimetric PM$_{2.5}$ mass concentrations can also be directly modeled. Firstly, a basic model regressing mean 48-h HAPEx signal on total integrated PM$_{2.5}$ mass concentrations was done. Then, a second regression was run, limiting the comparison to gravimetric filter samples containing less than 20% mass by dust ($n = 8$). The resulting relationship of this regression gives a better estimate of the effect primary biomass emissions have on 48-h mean HAPEx signal.

Lastly, to simulate a situation where gravimetric filter data are not available (excluding EC, OC, dust, etc.) but low-cost tools are, a stepwise multilinear regression was run, modeling the logarithm of total integrated PM$_{2.5}$ mass concentration with G-Pod and baseline-corrected HAPEx measurements coupled with "season", "LocationType" (e.g., urban/rural), and "CoverageClass" (i.e., kitchen classifications), resulting in Equation (7). Dependent variable normality was achieved by taking the logarithm of total PM$_{2.5}$ mass concentration reflecting a log-linear form. "PercbyMCE" is the fraction of summed HAPEx

readings corresponding to MCE > 0.995 to the total summation of HAPEx readings in a 48-h deployment, and it is a proxy for the fraction of readings associated with limited to no combustion activity.

$$
\begin{aligned}
\log([PM_{2.5}]) \sim{}& \beta_0 + \beta_1(mean\_temp) + \beta_2(mean\_RH) + \beta_3(percbyMCE) \\
& + \beta_4(meanHAPEx) + \beta_5(season) \\
& + \beta_6(CoverageClass \times LocationType) + \epsilon.
\end{aligned}
\tag{7}
$$

An analogous, but slightly variable model to Equation (7) was generated through stepwise regression using Wang et al.'s pointwise RH correction (Equation S1, Supplementary Materials) on baseline-corrected HAPEx readings and is described by Equation (8). The variable "CleanSD" is the average signal threshold of the paired monitors representing background signal noise for individual HAPEx.

$$
\begin{aligned}
\log([PM_{2.5}]) \sim{}& \beta_0 + \beta_1(mean\_temp) + \beta_2(mean\_RH) + \beta_3(percbyMCE) \\
& + \beta_4(meanHAPEx_{RH-corr}) + \beta_5(cleanSD) \\
& + \beta_6(mean_{CO2} \times season) + \beta_7(CoverageClass \times LocationType) \\
& + \varepsilon.
\end{aligned}
\tag{8}
$$

Data processing and analysis were executed using custom scripts in the Matlab environment (R2018b, Mathworks). The baseline correction algorithm was originally written in Python but later translated to Matlab for uniformity with subsequent analysis. Bland–Altman plots were made using Bland–Altman [51] and Lin's CCC calculated using a modified version of f_CCC [52]. Colored grouped boxplots were made using grammar of graphics plotting (GRAMM) [53]. Stepwise regression was done using Matlab's function stepwiselm. Reported coefficients of determination ($R^2$) are all adjusted for the number of model coefficients (adjusted $R^2$) and resulting degrees of freedom. A copy of the baseline algorithm Matlab script can be found in the Supplementary Materials.

## 3. Results

A total of 29 and 42 microenvironment samples were collected between September 2017 and March 2019 from a total of 28 rural and 32 urban households, respectively. Nine filter samples were omitted from the analysis due to insufficient gravimetric sampling time (<41.6 h, <87% of 48 h). The average filter sampling time was 47 h (SD = 1, $n = 62$). Deployments containing less than 95% matched minutes between at least one HAPEx and G-Pod measurement were also omitted from this analysis ($n = 18$), leaving 44 filter-to-HAPEx comparisons. Deployments with paired 48-h HAPEx readings disagreeing by more than three standard deviations of all pairs were flagged, and the lower of the two HAPEx values was used ($n = 1$). Out of the 71 microenvironment deployments, 13 had only one HAPEx unit operating properly. A total of 163,280 G-Pod minute measurements were collected, 143,318 of which had at least one HAPEx reading and 121,198 of which had paired HAPEx readings.

### 3.1. Cumulative Gravimetric and Carbonaceous PM_{2.5}

Mean 48-h kitchen area $PM_{2.5}$ concentrations were 168μg·m$^{-3}$ in rural homes ($n = 25$) and 116μg·m$^{-3}$ in urban homes ($n = 37$) (Figure 3). These integrated mass concentrations ranged from 53–354 μg·m$^{-3}$ in rural homes and 16–1114 μg·m$^{-3}$ in urban homes. Median concentrations were less than means at 149 μg·m$^{-3}$ and 70 μg·m$^{-3}$ in rural and urban homes, respectively.

EC and OC mass concentrations were 1.7 μg·m$^{-3}$ and 25.4 μg·m$^{-3}$ in rural kitchens, and 1.1 μg·m$^{-3}$ and 15.4 μg·m$^{-3}$ in urban kitchens, respectively. The ratio of EC to OC (EC/OC) was roughly equal in urban (0.08) and rural kitchens (0.07).

The sum of carbonaceous mass and non-carbonaceous organic mass (OM + EC) was on average 42 μg·m$^{-3}$ and 26 μg·m$^{-3}$ in rural and urban kitchens, respectively. This metric approximates the contribution of primary combustion emissions to total $PM_{2.5}$, and ranged from a negligible fraction

to virtually all PM$_{2.5}$ mass with a mean fraction of 30% (median = 29%) in rural kitchens and 40% (median = 22%) in urban kitchens.

Mean inorganic, non-carbonaceous mass concentrations (assumed to be primarily road and soil dust in this region) were 125 µg·m$^{-3}$ in rural homes and 90 µg·m$^{-3}$ in urban homes. These concentrations represent, on average, 71% and 61% of overall PM$_{2.5}$ mass in rural and urban homes respectively, yet this percentage ranged from near 0% to 99% in both settings. Ambient air quality measurements made on the outskirts of Navrongo from 2010 of PM$_{2.5}$ and PM$_{10}$ allowed Ofosu and colleagues to perform source apportionment finding resuspended road and soil dust to be the leading contribution (52%) to total PM$_{2.5}$ mass with biomass combustion as the second most important source (16%) [15].

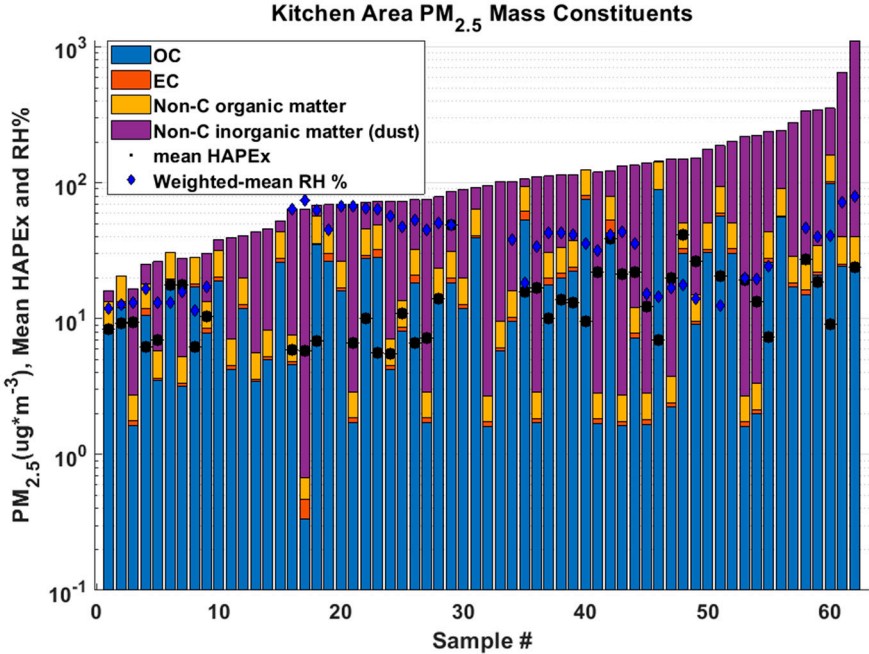

**Figure 3.** Breakdown of gravimetric 48-h particulate matter less than 2.5 microns in diameter (PM$_{2.5}$) mass concentrations by organic carbon (OC), elemental carbon (EC), non-carbonaceous organic matter, and non-carbonaceous inorganic matter with associated HAPEx-weighted mean relative humidity (RH, %) and mean baseline-corrected HAPEx signal. Note the logarithmic *y*-axis.

Sixty percent (37/62) of cumulative kitchen area 48-h total PM$_{2.5}$ concentrations were higher than the World Health Organization (WHO) interim target one (IT-1) 24-h average PM$_{2.5}$ mass concentration level of 75 µg·m$^{-3}$ [54] (Table 1). The fraction of 48-h samples exceeding this limit were not equal across urban and rural kitchens. In fact, 43% of urban kitchens exceeded the target level compared to 84% of rural kitchens. All rural kitchens surpassed the interim-2 target level of 50 µg·m$^{-3}$. It is important to note that these sample period exceedance percentages are quite conservative estimates of the percentage of days (24-h periods) exceeding the target levels.

When limiting the analysis to only non-carbonaceous, inorganic mass (dust), 71% (44/62) of samples surpassed the WHO guideline level of 25 µg·m$^{-3}$ (Table 2). In fact, 57% (21/37) of urban kitchens exceeded the guideline level relative to 92% (23/25) of rural kitchens. Clearly, dust is a major contributor to household PM$_{2.5}$, and efforts to reduce total PM mass concentrations in kitchens ought not to neglect baseline contributions from dust.

**Table 1.** World Health Organization (WHO) 24-h average total particulate matter less than 2.5 microns in diameter ($PM_{2.5}$) mass concentration targets and guidelines, and percentage of rural and urban 48-h kitchen samples exceeding each level. These percentages represent the lowest percentage of individual days exceeding each level. IT—interim target, AQG—air quality guideline.

| Criterion | $PM_{2.5} > 75$ μg·m$^{-3}$ | $PM_{2.5} > 50$ μg·m$^{-3}$ | $PM_{2.5} > 37.5$ μg·m$^{-3}$ | $PM_{2.5} > 25$ μg·m$^{-3}$ |
|---|---|---|---|---|
| WHO 24-h metric | IT-1 | IT-2 | IT-3 | AQG |
| Rural | 84% (21/25) | 100% (25/25) | 100% (25/25) | 100% (25/25) |
| Urban | 43% (16/37) | 62% (23/37) | 76% (28/37) | 92% (34/37) |
| Overall | 60% (37/62) | 77% (48/62) | 85% (53/62) | 95% (59/62) |

**Table 2.** World Health Organization 24-h average total $PM_{2.5}$ mass concentration targets and guidelines, and percentage of rural and urban 48-h kitchen samples exceeding each level for *dust concentration only*. These percentages represent the lowest percentage of individual days exceeding each level. IT—interim target, AQG—air quality guideline.

| Criterion | $PM_{2.5,dust} > 75$ μg·m$^{-3}$ | $PM_{2.5,dust} > 50$ μg·m$^{-3}$ | $PM_{2.5,dust} > 37.5$ μg·m$^{-3}$ | $PM_{2.5,dust} > 25$ μg·m$^{-3}$ |
|---|---|---|---|---|
| WHO 24-h metric | IT-1 | IT-2 | IT-3 | AQG |
| Rural | 68% (17/25) | 84% (21/25) | 88% (22/25) | 92% (23/25) |
| Urban | 27% (10/37) | 41% (15/37) | 49% (18/37) | 57% (21/37) |
| Overall | 44% (27/62) | 58% (36/62) | 65% (40/62) | 71% (44/62) |

*3.2. Low-Cost Monitor Performance and Baseline Signal Correction*

3.2.1. High Temporal Reproducibility: Base and Threshold Signal Filters

The base filter detected, flagged, and omitted 409 of 121,198 paired baseline-corrected minute anomalies (0.34%) and 400 of 121,198 paired raw minute anomalies (0.33%) from the analysis. The omission criteria for the base filter (3 × SD of 5-min rolling differences) were 364 and 755 for baseline-corrected and raw readings, respectively. On median, paired 5-min rolling average differences (12.8) were nearly five times higher than those of baseline-corrected readings (2.6). This is the first line of evidence of a functioning baseline drift correction algorithm.

Although squared Pearson's $r$ values were not significantly different ($r^2 = 0.47$, $p = 0.31$) between baseline-corrected and raw 1-min values using Fisher's $r$ to $z$ transformations, the baseline-corrected coefficient of reproducibility (RPC = 1.96 × SD of differences) was almost half that of the raw RPC, showing greater reproducibility of baseline-corrected readings (Figure 4). Lin's concordance correlation coefficients (CCC) were no different between raw and baseline-corrected 1-min data at the 5% significance level ($CCC_{bcor} = 0.687$, 95% confidence interval (CI): 0.685–0.689; and $CCC_{raw} = 0.684$, 0.681–0.686) (Figure 4).

When 5-min rolling averages were taken, baseline-corrected and raw paired correlations increased substantially, yet baseline-corrected correlation ($r^2 = 0.84$, 95% CI: 0.846–843) was significantly ($p = 0$) higher than the raw ($r^2 = 0.60$, 0.599–0.607). RPCs dropped considerably for both sets of data, yet the decrease was more pronounced for baseline-corrected pairs (Figure 5). Spearman's rho is useful for characterizing non-parametric associations between measures and followed similar trends as Pearson's $r$ due to the overall linearity between HAPEx readings. Perhaps most striking, the gap between CCC for corrected (0.917, 95% CI: 0.916–0.918) and raw (0.758, 0.756–0.760) data grew considerably, further affirming the value of the drift correction algorithm as significantly and substantially improving agreement between paired readings when a 5-min rolling average is taken (Figure 5). False signals indicative of baseline drift are likely to affect paired monitor signals due to sheer proximity to one another, potentially preserving Pearson's correlation, yet the magnitude of the drift is likely to vary as well, and this discrepancy is captured by the RPC and CCC.

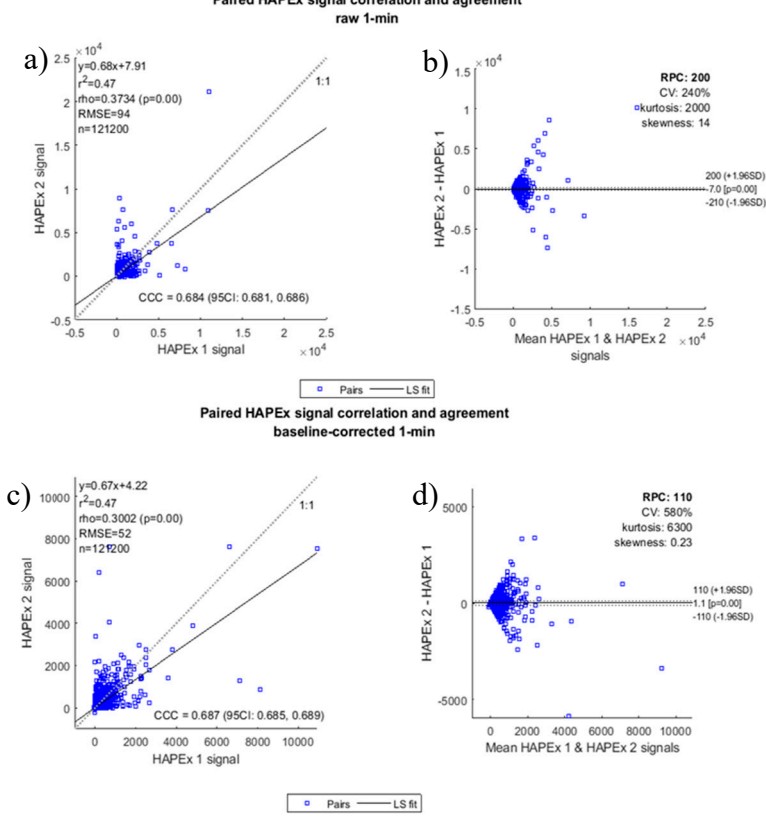

**Figure 4.** Scatterplots (**a**,**c**) and Bland–Altman plots (**b**,**d**) of paired 1-min HAPEx signals for raw (**a**,**b**) and baseline-corrected (**c**,**d**) data. Pearson's correlation (*r*), Lin's concordance correlation coefficients (CCC) with 95% confidence intervals (95CI), a linear least squares fit (LS, solid line) with root mean squared error (RMSE), coefficients of variation (CV) and coefficients of reproducibility (RPC) are reported.

The threshold filter identified and replaced 74,857 of 238,686 raw minute readings (31%) and 171,607 of 238,675 baseline-corrected minute readings (72%) with a monitor-deployment-specific threshold signal. A greater percentage of baseline-corrected data were replaced due to the fact that raw readings typically drifted upward, reducing the number of readings recorded below the threshold signal (TS). Baseline-corrected and raw monitor-deployment-specific TS followed a log-normal distribution and ranged from 5.4 (determined through zeroing) to 22 with a mean of 6.6.

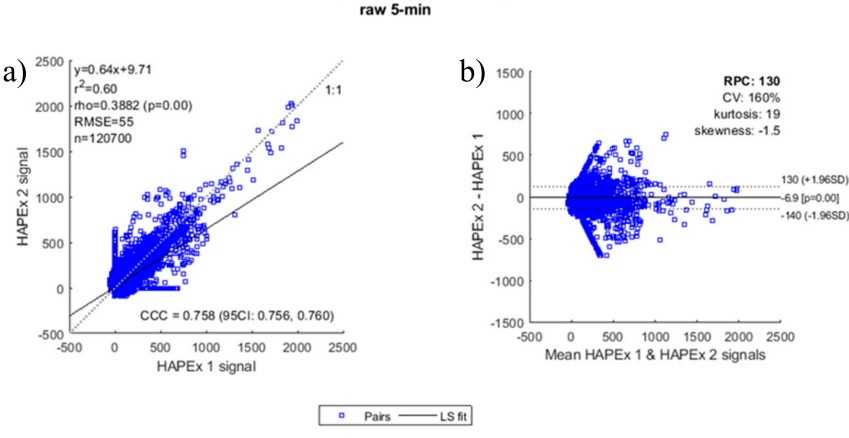

**Figure 5.** *Cont.*

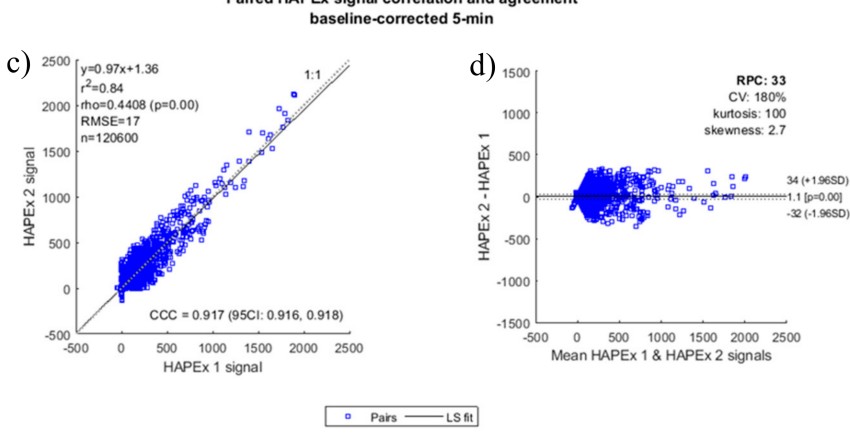

**Figure 5.** Scatterplots (**a**,**c**) and Bland–Altman plots (**b**,**d**) of paired, 5-min rolling averaged HAPEx signals for raw (**a**,**b**) and baseline-corrected (**c**,**d**) data. Pearson's correlation (*r*), Lin's concordance correlation coefficient (CCC) with 95% confidence intervals (95CI), a linear least squares fit (LS, solid line) with root mean squared error (RMSE), coefficients of variation (CV) and coefficients of reproducibility (RPC) are reported.

### 3.2.2. Lower Temporal Reproducibility: Paired Mean HAPEx 48-h Agreement

Paired 48-h mean HAPEx readings (with base and threshold filters applied) were compared using raw (*n* = 39) and baseline-corrected (*n* = 40) values (Figure 6) and colored by HAPEx weighted-mean RH. Coefficients of determination (adjusted $R^2$) and concordance correlation coefficients (CCC) were calculated for each set of measurements. Firstly, the CCC was significantly higher (*p* < 0.05) for baseline-corrected readings (0.96, 95% CI: 0.94–0.98) compared to raw counterparts (0.78, 0.68–0.85). Baseline-corrected mean 48-h precision was substantially improved over raw readings, further affirming the baseline drift algorithm. The tendency for raw measurements to drift upward giving false positives is illustrated in the range of 48-h raw mean readings (5.4–496) compared to the drastically lower baseline-corrected range (5.4–48). No significant relative humidity effect on precision was discovered, although absolute differences between paired mean 48-h readings increased as RH increased (Figure S3, Supplementary Materials). Also, coefficients of determination did not change for raw ($R^2$ = 0.67) data and actually lowered for baseline-corrected ($R^2$ = 0.85) data using 48-h geometric mean values.

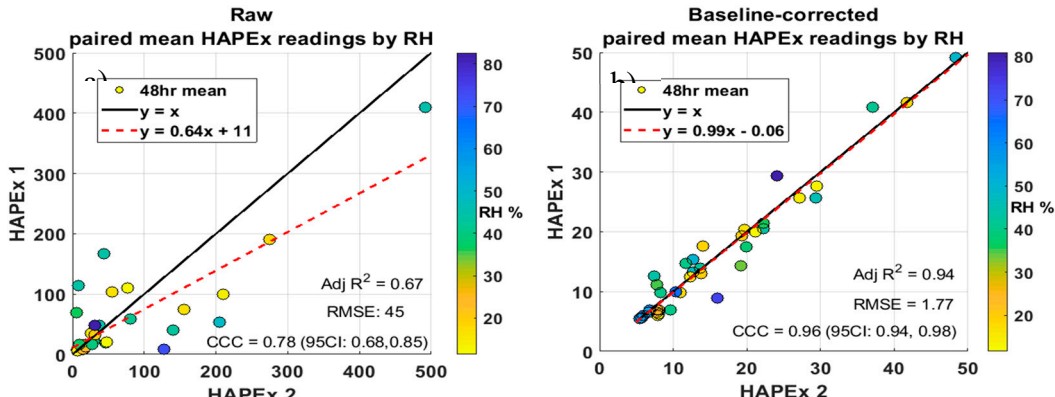

**Figure 6.** The 48-h mean paired HAPEx readings from (**a**) raw and (**b**) baseline-corrected data colored by HAPEx-weighted mean RH (%). Coefficients of determination ($R^2$), root-mean-squared error (RMSE) and Lin's concordance correlation coefficients (CCC) with 95% confidence interval (95CI) are reported.

### 3.3. Encompassing Environmental Variability in Kitchen Areas toward Understanding Effects on Low-Cost Sensor Readings

Figure 7 shows mean (95% CI via bootstrapping) kitchen environmental measurements collected by the G-Pod by hour of day and by season. Kitchen temperature, for example, varies more by hour of

the day than seasonally, whereas relative humidity varies more by season than diurnally. The degree to which this variability is encompassed in 48-h mean metrics is examined in the section modeling particle coefficients. Unsurprisingly, $CO_2$ baseline concentrations vary substantially by season, driven by vegetation and climate patterns. Additionally, peaks in CO concentrations vary dramatically by hour of day and to a lesser extent seasonally, as found before in a previous cookstove study in the region [38,41]. Background-subtracted CO and $CO_2$ concentrations used in estimating MCE during a 48-h deployment remove the seasonal trends but preserve hour-of-day trends, clearly showing depressions from unity (an MCE of ~1 signifies no combustion) corresponding to combustion activity at typical mealtimes (Figure S4, Supplementary Materials). It is important to point out that rural and urban deployments did not fully cover all seasons, resulting in confounding between season and urban/rural variables.

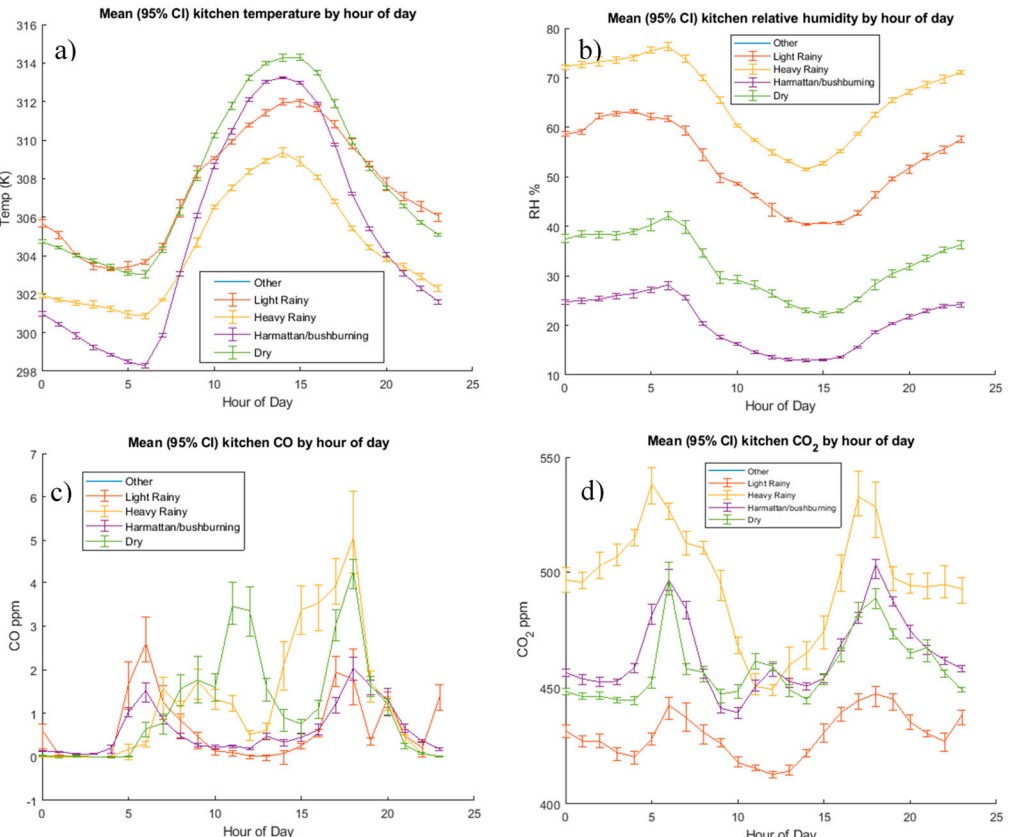

**Figure 7.** Time-of-day plots marking mean (95% CI) (**a**) temperature (K), (**b**) relative humidity (%), (**c**) CO concentration (ppm), and (**d**) $CO_2$ concentration (ppm) G-Pod measures in study kitchens by hour of the day and colored by season. "Other" is a transitional season of two weeks between "dry" and "light rainy" seasons.

Figure 8 illustrates the difference between means and HAPEx-weighted means for an overall relative humidity value over a 48-h period. Sixteen HAPEx monitor deployments had at least a 10% relative difference between the 48-h RH mean and weighted mean. Similar findings were discovered for other covariates (i.e., CO, $CO_2$, and temperature).

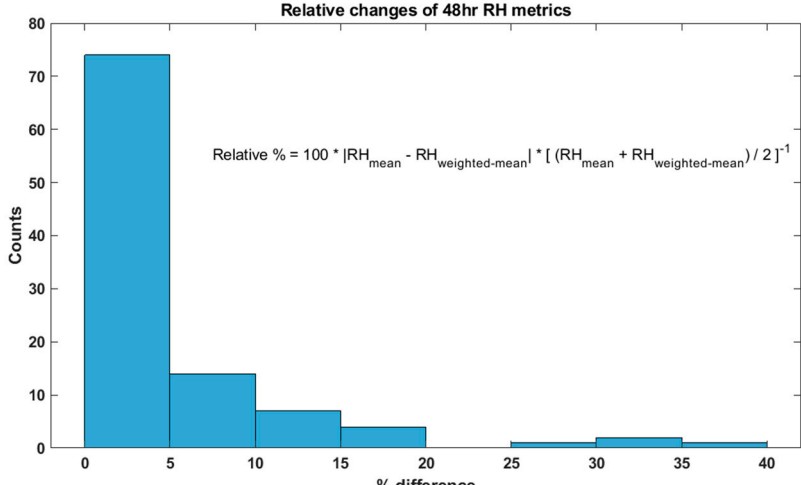

**Figure 8.** Relative differences between 48-h mean and HAPEx-weighted mean relative humidity.

*3.4. Baseline Correction Quality Check: Low-Cost Optical Sensor vs. Gravimetric Filter*

Regressions of 48-h mean HAPEx readings on integrated $PM_{2.5}$ mass constituents show interesting trends. Firstly, Figure 9 illustrates the relationship between (a) raw and (b) baseline-corrected 48-h mean HAPEx readings and total $PM_{2.5}$ mass concentration ($\mu g \cdot m^{-3}$). The baseline-corrected data mean HAPEx signal ($R^2 = 0.48$) explains more than twice as much variation in total integrated filter mass concentration as the raw data ($R^2 = 0.22$). For baseline-corrected HAPEx signal, both the intercept (5.0, 95% CI: 1.9–8.8) or modeled HAPEx signal in clean air and slope (0.089, 95CI: 0.061–0.117) or particle coefficient are significant ($p < 0.05$). For raw HAPEx data, the slope is substantially higher (0.479). Points are colored by EC/OC, with higher values corresponding to darker shades. Elemental carbon absorbs more light relative to OC and would in theory result in less scattered light available for the photodiode to detect. Black carbon (BC), a close counterpart to EC, is commonly measured optically using light absorptivity principles. When the regression is limited to gravimetric samples containing less than 20% mass of dust (Figure 10), the slope increases slightly to 0.112 (0.067–0.157, $p < 0.05$) and 84% of mean HAPEx variance is explained by total $PM_{2.5}$ mass concentration. This slope reflects HAPEx sensitivity to primary biomass emissions. Regressions of mean HAPEx readings on EC, OC, or TC show poor correlation, whereas dust mass alone explains 46% of HAPEx variance. Moreover, regressions grouped by urban/rural and season were explored, showing variability in slopes and intercepts across those dimensions (Figure S5, Supplementary Materials).

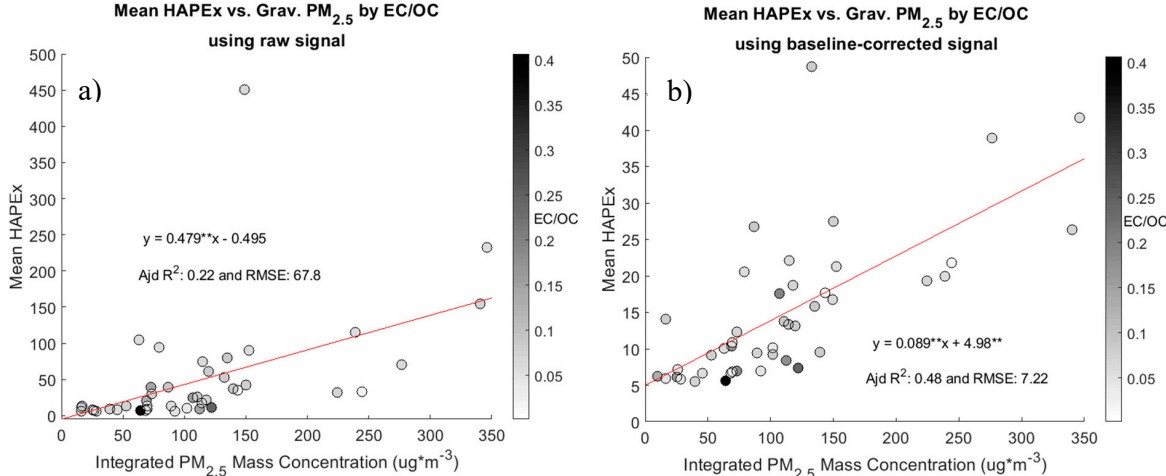

**Figure 9.** Kitchen area 48-h mean HAPEx readings regressed against gravimetric total PM$_{2.5}$ mass concentration colored by the ratio of elemental to organic carbon (EC/OC) for (**a**) raw HAPEx signal and (**b**) baseline-corrected HAPEx signal (** $p < 0.01$).

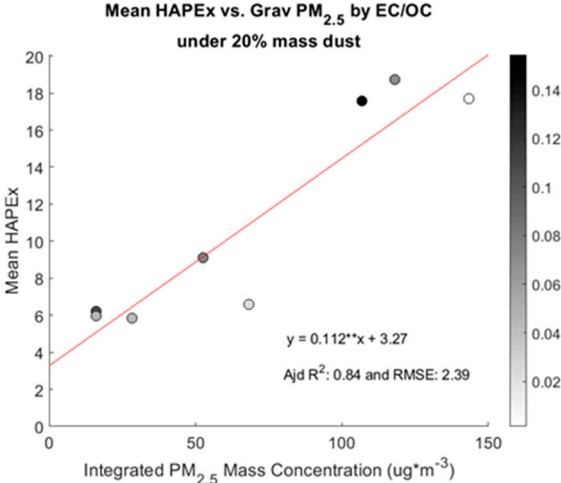

**Figure 10.** Kitchen area 48-h mean HAPEx readings regressed against gravimetric total PM$_{2.5}$ mass concentration colored by the ratio of elemental to organic carbon (EC/OC) for a subset of samples ($n = 8$) with less than 20% of total mass dust (** $p < 0.01$).

*3.5. Applying Pointwise RH Corrections from Literature*

Relative humidity corrections made to individual 1-min baseline-corrected readings improved linear relationships between mean HAPEx and gravimetric PM$_{2.5}$ mass concentrations (Figure 11). Corrections made from Wang et al. explained an additional 11% variance in mean HAPEx readings ($R^2 = 0.59$) whereas the correction characterized by Chakrabarti et al. explained and additional 6% variance ($R^2 = 0.55$). Both corrections lowered the modeled intercept (3.93, $p < 0.05$ and 4.19, $p < 0.05$), whereas the correction derived from Wang et al. resulted in a higher particle coefficient (0.104, $p < 0.05$) as opposed to Chakrabarti et al.'s which remained similar to the uncorrected data (0.086, $p < 0.05$). Both RH-corrected slopes (PCs) are within the 95% CI of the uncorrected data slope.

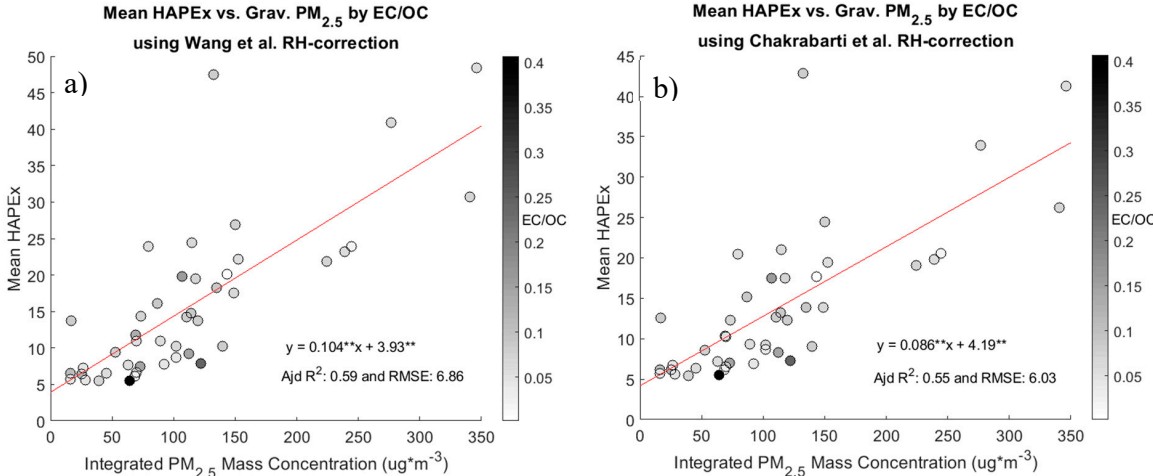

**Figure 11.** Kitchen area 48-h mean HAPEx readings regressed against gravimetric PM$_{2.5}$ mass concentrations using pointwise RH corrections derived from (**a**) Wang et al., 2015 and (**b**) Chakrabarti et al., 2004. Points are colored by EC/OC (** $p < 0.05$).

### 3.6. Low-Cost Sensor's Ability to Predict Gravimetric Filter Measures

Part of evaluating the low-cost optical sensor was modeling the predictive capacity of the low-cost readings to estimate reference measurements. Model results estimating particle coefficients and total PM$_{2.5}$ mass concentrations are described below. Previous literature focused on modeling particle coefficients. We explored the utility of an alternative approach where we directly model PM$_{2.5}$ mass concentration, a relationship by which the Gold Standard can be compared.

#### 3.6.1. Modeling Particle Coefficient

Recall particulate matter varies in composition, size, reflectivity, etc., causing variation in resulting particle coefficients (PCs). Particle coefficients were calculated from 48-h integrated gravimetric PM$_{2.5}$ mass concentrations and 48-h mean HAPEx baseline-corrected values using Equation (1), and then they were modeled using environmental factors (e.g., temperature, relative humidity, dust concentrations, etc.) and classifications (e.g., urban/rural and season) as predictors to understand what drives variation in PCs (Equations (2)–(6)). The mean and median rural particle coefficients (PCs) were 0.143 (±0.064, $n = 18$) and 0.130, respectively, ranging from 0.060 to 0.308. The mean and median urban PCs were 0.187 (±0.163, $n = 26$) and 0.132, respectively, ranging from 0.068 to 0.845. Figure 12 shows PCs paired with gravimetric mass concentrations and weighted-mean RH on a logarithmic vertical axis.

Mean RH and HAPEx-weighted mean RH alone were poor predictors of PC, explaining negligible (≤5%) variation (Table 3, adjusted $R^2$). Very weak correlation ($R^2 = 0.04$) between 48-h mean RH values and PC was found with no significant linear trend ($p = 0.11$) (Figure S6, Supplementary Materials). However, the multilinear model (Equation (6)) describes 79% of PC variation with an RMSE of 0.06 and normally distributed residuals (Figure S8, Supplementary Materials). Interactions of season and urban/rural (LocationType) with mean HAPEx signal helped disentangle the effects those two variables had on regression slopes (PC), as illustrated in Figure S5 of Supplementary Materials, and quantified in Table 3. Dust concentrations interacting with season helped explain additional variability as the fraction of dust making up total PM$_{2.5}$ mass in kitchens varied seasonally (Figure S7, Supplementary Materials). Full model results can be found in the Supplementary Materials.

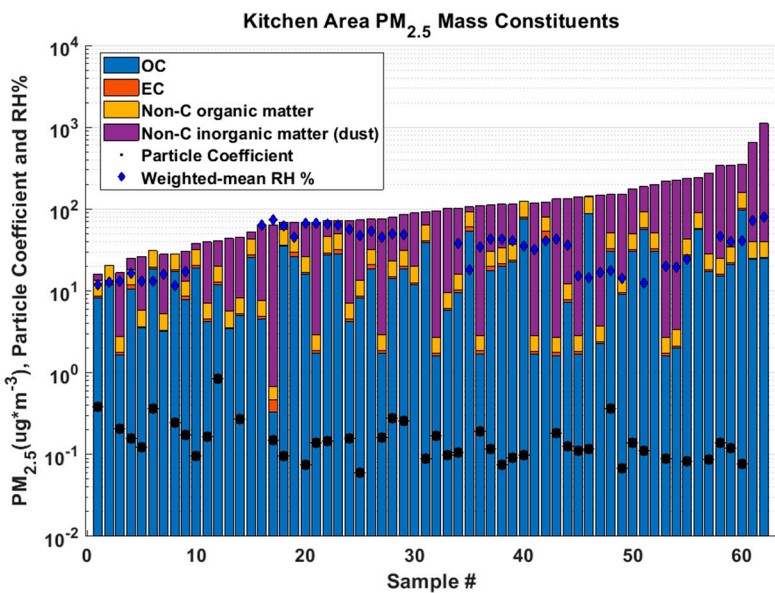

**Figure 12.** Breakdown of gravimetric 48-h PM$_{2.5}$ mass concentrations by organic carbon (OC), elemental carbon (EC), non-carbonaceous organic matter, and non-carbonaceous inorganic matter with associated HAPEx-weighted mean RH (%) and particle coefficients. Note the logarithmic *y*-axis.

**Table 3.** Regression results modeling particle coefficients (PCs) (*n* = 44) with relative humidity (RH) and other variables. RMSE—root-mean-squared error; meanHAPEx—PM monitor mean 48-h signal.

| Model Coefficients | b(0), a | b(1), b | b(2) | b(3) | p | Adj. $R^2$ | RMSE |
|---|---|---|---|---|---|---|---|
| **Equation (2) (mean RH)** | 0.117 * | 0.001 | - | - | - | 0.02 | 0.13 |
| **Equation (3) (HAPEx-weighted RH)** | 0.115 * | 0.002 | - | - | - | 0.03 | 0.13 |
| **Equation (4) (logarithmic)** | 3.75 * | −0.18 | - | - | - | 0.05 | 0.36 |
| **Equation (5) (nonlinear)** | 0.153 * | 0.037 | - | - | - | 0.007 | 0.13 |
| **Equation (6) (multilinear)** | 0.112 | - | - | - | **0.005** | | |
| meanHAPEx | 0.007 | - | - | - | 0.09 | | |
| dust_concentration | $−9 \times 10^{-4}$ | - | - | - | 0.10 | | |
| Location_Type: rural | Reference Group | | | | | | |
| Location_Type: urban | 0.12 | - | - | - | 0.03 | | |
| meanHAPEx: season_Dry | Reference Group | | | | | | |
| meanHAPEx: season_Harmattan_bushburning | - | −0.001 | - | - | 0.80 | | |
| meanHAPEx: season_Heavy_Rainy | - | 0.005 | - | - | 0.33 | 0.79 | 0.061 |
| meanHAPEx: season_Light_Rainy | - | 0.07 | - | - | **0.00** | | |
| meanHAPEx: season_other | - | 0 | - | - | NA | | |
| dust_concentration: season_Dry | Reference Group | | | | | | |
| dust_concentration: season_Harmattan_bushburning | - | - | 0.0002 | - | 0.71 | | |
| dust_concentration: season_Heavy_Rainy | - | - | −0.001 | - | 0.17 | | |
| dust_concentration: season_Light_Rainy | - | - | −0.022 | - | **0.00** | | |
| dust_concentration: season_other | - | - | 0 | - | NA | | |
| meanHAPEx: LocationType_Rural | Reference Group | | | | | | |
| meanHAPEx: LocationType_Urban | - | - | - | −0.01 | **0.01** | | |

Note: * *p* < 0.05, else see *p* for Equation (6); NA—not available.

### 3.6.2. Modeling Gravimetric PM$_{2.5}$ Concentration

PM$_{2.5}$ mass concentrations were measured; however, modeling these measurements using HAPEx readings and other environmental covariates (e.g., temperature, relative humidity, season, kitchen characteristics, etc.) as predictors (as if filter measurements were not taken) allowed us to better understand the predictive capacity of the sensor when baseline-corrected HAPEx signal is not corrected for RH (Equation (7)) and when an RH correction is applied (Equation (8)). Modeling the log of 48-h integrated total PM$_{2.5}$ mass concentration using Equation (7) (see Supplementary Materials for full results) resulted in moderately high correlation ($R^2$ = 0.69, Figure 13a) and normally distributed residuals (Figure S9, Supplementary Materials). Predictor terms with significant coefficients ($p < 0.05$) included the overall intercept (β = 36.6, ±11.7 standard error), mean temperature (−0.1 ± 0.04), mean RH (−0.04 ± 0.01), meanHAPEx signal (0.046 ± 0.009), kitchens with roofs but no walls (−1.94 ± 0.75) relative to no roof with one wall (reference group), the "Harmattan/bushburning" (−0.59 ± 0.22) and "heavy rainy" (1.35 ± 0.57) seasons relative to the "dry" (reference group) and an interaction term of kitchens with a roof and four walls in the urban setting (−2.51 ± 0.82) relative to rural kitchens with no roof and one wall (reference group).

Results from modeling total PM$_{2.5}$ mass concentration using Equation (8) (see Supplementary Materials for full results) had an improved fit ($R^2$ of 0.79, Figure 13b) and a lower RMSE (0.35) than Equation (7) ($R^2$ of 0.69, RMSE of 0.42) also with normally distributed residuals (Figure S10, Supplementary Materials). Similar to results from Equation (7), predictor terms with significant coefficients ($p < 0.05$) included the overall intercept (β = 41.4, ±11.96 standard error), mean temperature (−0.1 ± 0.04), mean RH (−0.03 ± 0.01), mean CO$_2$ (−0.01 ± 0.004), percbyMCE (−0.8 ± 0.3), meanHAPEx baseline-corrected signal (0.076 ± 0.01), kitchens with no roof or walls (2.50 ± 0.68), with roof and one wall (2.50 ± 0.69), with roof and four walls (1.28 ± 0.50), and with roof and no walls (2.1 ± 0.84) compared to kitchens with no roof and one wall (reference group), the "light rainy" (30.2 ± 11.9) season relative to the "dry" (reference group), urban kitchens (−2.4 ± 0.59) relative to rural kitchens (reference group), the HAPEx signal threshold ("cleanSD", −0.08 ± 0.03) and two interaction terms; one term for mean CO$_2$ and the "light rainy" season (−0.07 ± 0.27) relative to mean CO$_2$ and the "dry" season (reference group), and a second term for urban kitchens with no roof and two walls (3.36 ± 0.80) relative to rural kitchens with no roof and one wall (reference group). The "Harmattan/bushburning" season (−4.17 ± 1.93) relative to the "dry" season (reference group) had a *p*-value of 0.056.

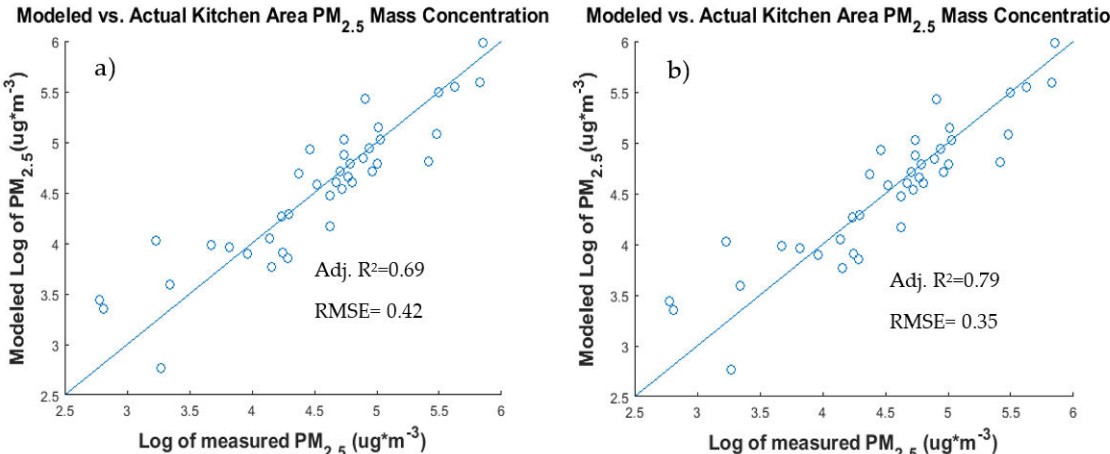

**Figure 13.** Modeled log of 48-h integrated PM$_{2.5}$ mass concentrations vs. log measured PM$_{2.5}$ mass concentrations using (**a**) Equation (7) without a pointwise RH correction, and (**b**) Equation (8) with pointwise RH-corrected data. Full model results can be found in the Supplementary Materials.

## 4. Discussion

This paper compares PM$_{2.5}$ concentration readings from HAPEx low-cost optical sensors and gravimetric filter samples in household kitchen areas in northern Ghana. In addition to cost savings, optical sensors provide time-resolved measurements, which can be important in understanding PM sources and trends. We note a few key methodological findings related to HAPEx signal processing, reproducibility at varying temporal scales, and comparisons to gravimetric filter results. However, we contextualize these methodological insights by discussing some substantive results from this field study. Analysis of findings specific to the Prices, Peers, and Perceptions (P3) study activities are planned for future work where, for example, we aim to investigate differences among study arms incorporating behavior (i.e., stove use, adoption results, time activity).

Firstly, it is worth noting the large contribution of dust mass to total kitchen area 48-h integrated gravimetric PM$_{2.5}$ mass in this field study. The average rural kitchen PM$_{2.5}$ mass is 71% dust and the average urban kitchen is 61% dust. These dust fractions are considerably larger than the estimated 52% contribution to ambient PM$_{2.5}$ measured in 2010 in Navrongo, the district capital and home to many of the urban sample households and within tens of kilometers from many of the rural sample households [15]. The increased dust fraction is likely linked to seasonal trends (Figure S6, Supplementary Materials) but also driven by human behavior. Anecdotal evidence suggests that many cooks and children prepare kitchens for cooking by sweeping the ground floors and often resuspending soil dust in the process—an activity commonly occurring directly prior to cooking, making high temporal measurements of exposure and behavior critical [9]. Interventions aimed at reducing total PM$_{2.5}$ mass concentrations to WHO interim target levels (I, II, and III) ought to consider a multifaceted approach targeting solid biomass combustion activity, as well as resuspended road and soil dust in some study regions. Even if EC and OM PM$_{2.5}$ mass constituents were reduced to zero, 68% and 27% of rural and urban kitchens in this study region exceeded the WHO interim target (IT-1) of 75 µg·m$^{-3}$. Long tails in dust particle size distributions could explain the large fraction of dust measured in the fine mode (PM$_{2.5}$), with some research suggesting a limited dependence of the particle size distribution of <5-µg dust on soil properties [55], motivating additional sampling.

A major methodological finding stemmed from firstly identifying baseline drift and then implementing and evaluating the baseline drift correction algorithm. Initially, the HAPEx sensor baseline drift presented a large challenge in a very specific regard. Specifically, by choosing to correct the baseline drift and thereby removing any possible long term (>80 min) background trend in signal responses to PM$_{2.5}$ concentrations, the HAPEx signal is effectively an indicator of elevations above those background trends—aligning well with our primary research objectives. However, the background trends are captured in the integrated gravimetric measurement resulting in positive overall intercepts of Equations (7) and (8) modeling total PM$_{2.5}$ mass concentrations. That being said, improvements from correcting the baseline drift (increased Pearson's and Lin's correlations and decreased RPC) resulted in significantly higher precision among paired HAPEx monitors at 5-min rolling ($R^2$ = 0.60 to 0.84) and 48-h ($R^2$ = 0.67 to 0.94) averaging times, making relative comparisons among loggers reliable and achievable. Notably, at 1-min resolution, the impact of the baseline-correction on Pearson's *r* or Lin's CCC was not significant, yet the reproducibility coefficient of the baseline-corrected data was lower, suggesting improvements even at very high temporal resolution. Moreover, the baseline drift correction substantially improved relationships between 48-h mean HAPEx signal and total 48-h integrated gravimetric PM$_{2.5}$ mass concentration, explaining more than twice as much variation in 48-h mean baseline-corrected signal compared to raw signal.

Particle chemistry (e.g., size, composition, reflectivity, shape, etc.) impacts optical sensor responses causing variation in readings. Measurements of EC and OC mass (as well as ratios of EC/OC) help describe the chemical make-up of particulates. EC and OC kitchen mass concentrations and resulting EC/OC ratios were similar to kitchen measurements made in other households in the KN districts 3–5 years ago during the Research of Emissions Air Quality, Climate and Cooking Technologies in Northern Ghana REACCTING cookstove study [38], although unfortunately total PM$_{2.5}$ mass measurements were not

collected in that study with which to compare. Correlations between mean HAPEx signal and EC, OC mass concentrations and EC/OC were surprisingly low, as the large variability in HAPEx signal was mostly explained by dust concentrations. However, when constraining the analysis based on particle properties to a subset ($n = 8$) of samples with less than 20% mass as dust, TC (EC + OC) explained 88% of the variance in mean HAPEx signal, with OC having high correlation ($r = 0.92$) with meanHAPEx signal.

As anticipated, relative humidity effects on HAPEx signal were found. Firstly, particle coefficient variation was not well explained by 48-h RH metrics. This is most likely the result of 48-h variability that cannot be encompassed by mean or HAPEx-weighted mean RH. Rather, correcting pointwise HAPEx signal by RH was more successful. Pointwise RH corrections increased correlation between mean 48-h HAPEx signal and total $PM_{2.5}$ mass concentrations, and reduced mean 48-h absolute differences between paired HAPEx monitors while also reducing trends in those paired differences by RH (Figure S3, Supplementary Materials).

Particle coefficients and total $PM_{2.5}$ mass concentrations varied by location and season, yet HAPEx sensor readings contributed important information in explaining variation in these measures, demonstrating their utility in this context.

Firstly, in this study, modeled PCs for urban kitchens were significantly higher (~2×) than rural kitchens, most likely caused by varying PM mixtures emitted from cooking fuels, as the urban population uses more LPG and charcoal [14,34,40]. However, this variability was also explained by interactions of dust concentrations and season. Seasonal variation of dust mass fraction, notably in the "light rainy" season relative to the "dry" season explains variability in PCs.

For comparison, particle coefficients determined for HAPEx monitors deployed in Indian homes burning biomass as part of NexLeaf's low-cost PM assessment averaged to 0.223 (4.48 $PM_{2.5}$ µg·m$^{-3}$/HAPEx reading), slightly higher than those found here [23]. Local particle chemistry and environmental factors could explain this difference. However, they did not correct HAPEx signal for baseline drift and, therefore, their calculated PCs would be higher (higher overall mean HAPEx readings and, therefore, higher PC). Interestingly, higher agreement was found when comparing our PCs to those they determined in a laboratory setting exposing HAPEx to woodsmoke (PC = 0.176, or 5.68 $PM_{2.5}$ µg·m$^{-3}$/HAPEx reading). Perhaps this is less surprising when considering there may have been less baseline drift in the laboratory tests than the field.

In a pilot setting, HAPEx monitors were also deployed in Guatemalan kitchens alongside pump-and-filter measurements of $PM_{2.5}$ to measure cooking emissions from 10 gasifier cookstoves and 10 traditional fires, finding particle coefficients ranging between 0.4 and 0.5 [21]. Again, these PCs have higher agreement with those from this study when not correcting for baseline drift (PC = 0.479, Figure 9a).

Secondly, the HAPEx sensor provided significant explanatory information when modeling total $PM_{2.5}$ mass concentrations with only measurements from the microenvironment apparatus. Integrated $PM_{2.5}$ mass concentrations were substantially lower in urban kitchens and substantially higher in the "light rainy" season relative to rural kitchens and the "dry" season, respectively. Kitchen characteristics play a role in total $PM_{2.5}$ mass concentrations with mixed findings. Kitchens with no roof or walls, a roof and one wall, roof with four walls, or roof with no walls have higher concentrations than kitchens with no roof and one wall. Roofs tend to be associated with higher concentrations, which intuitively makes sense as ventilation is restricted by roofs, except oddly when the kitchen is completely outdoors with no nearby structures (no roof or walls). Higher HAPEx-weighted mean temperature and RH are both associated with lower concentrations. One notable difference between the model incorporating the pointwise RH correction from Wang et al. (Equation (8)) and the model with no pointwise RH correction (Equation (7)) is the significance of the percentage of summed HAPEx readings recorded (percbyMCE) when modified combustion efficiency (MCE) was near background (>0.995), indicative of limited to no combustion activity. In Equation (8), this metric is a significant predictor ($p < 0.05$) of lower mean 48-h $PM_{2.5}$ concentration, whereas it is not a significant factor in Equation (7). Without knowing the exact explanation for this difference, perhaps the pointwise RH correction

reduces measurement error in high-humidity environments, resulting in the percbyMCE explaining more variation in $PM_{2.5}$. In summary, when incorporating variability explained by covariates such as season and location (i.e., urban/rural) with RH-corrected and baseline-corrected HAPEx signal, overall correlations with gravimetric $PM_{2.5}$ meet the Gold Standard criterion.

*Limitations and Opportunities for Improvement*

Although reported as most sensitive to PM with aerodynamic diameter between 1 and 3 μm, the HAPEx Nano is a passive monitor with no size selection inlet; therefore, it does not specifically discern between fine ($PM_{2.5}$) and coarse ($PM_{2.5-10}$) particulates. Developments are underway to make improvements to subsequent versions of the HAPEx Nano including on-the-fly zeroing, forced flow, and inlet size selection options [56].

The P3 air quality measurements campaign also included stove use monitoring on all improved and traditional stoves, proximity activity monitoring [9], personal exposure monitoring, and surveying in households enlisted in the air quality subset, including all the household samples reported here. These extra data streams can be layered over kitchen-level measurements to inform time-resolved HAPEx signal further (e.g., what stove and fuels were used); however, that was beyond the scope of this paper. Cross-validations of PC and total $PM_{2.5}$ concentration models, as pursued recently by Tryner and co-workers in explaining variation in PCs calculated from mid/high-range personal $PM_{2.5}$ exposure monitors [12], were not possible given the limited number of samples. There is a need for additional collocations of low-cost sensors with direct measurements of $PM_{2.5}$ mass concentrations at high temporal resolution in the field.

Issues encountered with the HAPEx Nano included (1) programming incorrect instrument settings resulting in logging all zeros ($n = 10$ deployments or 28,800 min; fortunately, many were paired with another working monitor); (2) battery failure that went unnoticed due to a lack of clear indicators ($n = 3-5$ deployments); (3) firmware settings with not a number ("NaN") values logged below $-99$ ($n = 2$ deployments or 5680 min); and (3) connectivity issues between HAPEx monitors and the computer, most likely caused by corrosion of the USB terminal by moisture over the two-year sampling campaign (8–15 HAPEx monitors). Others reported failure rates >40% in the field, which is still a major challenge facing the uptake of low-cost tools [20]. Duplicate measurements are highly recommended. However, it is important to note that the HAPEx signal is a product of on-board, proprietary firmware algorithms developed by the manufacturer (Climate Solutions Consulting) for the Sharp GP2Y1010, whereas our signal processing is conducted on raw 1-min outputs from averaged 20-s HAPEx Nano readings; therefore, it is directly applicable to others using the HAPEx Nano (v.3).

## 5. Conclusions

A low-cost (159 USD), passive, optical PM tool, called the HAPEx Nano, was evaluated in 60 Ghanaian kitchens primarily burning solid biomass and some using liquid propane gas fuels, representing a real-world field setting. The low-cost PM tool was collocated with high temporal measurements of temperature, relative humidity, and gaseous combustion emissions (e.g., CO and $CO_2$), as well as gravimetric pump and filters used to measure 48-h integrated $PM_{2.5}$ mass concentrations from a total of 71 kitchen deployments. The 48-h integrated total $PM_{2.5}$, elemental carbon, organic carbon, non-carbonaceous organic mass, and non-carbonaceous inorganic mass concentrations were reported for 62 kitchen areas, showing unexpectedly large contributions from dust, as well as primary biomass aerosols. A large percentage of measured total $PM_{2.5}$ mass concentrations, varying across urban and rural locations, surpassed the World Health Organization interim targets and air quality guideline, even for dust concentrations alone. A HAPEx signal baseline algorithm was introduced to correct for large drift in baseline signal and assessed by comparing baseline-corrected and raw paired readings at variable time resolution (1-min, 5-min, and 48-h), showing significant improvements of agreement and reproducibility and increased precision at 5-min rolling and 48-h averaging times; however, these improvements were limited to only increased reproducibility at 1-min readings, suggesting

more effectiveness at larger time averaging intervals. Relative humidity corrections found in existing optical PM sensing literature were applied to 1-min and 48-h mean HAPEx signals. Pointwise (1-min) RH corrections explained more variation in particle coefficients (48-h mean optical-to-gravimetric ratio) than 48-h mean RH or 48-h HAPEx-signal-weighted mean RH. Sampled kitchen environments are highly variable with respect to RH, temperature, and gaseous combustion pollutants across 48-h deployment periods and across seasons, making low-cost optical measurements more challenging to understand; however, when incorporated in multilinear regression models, they yield valuable information, resulting in correlations between 48-h mean HAPEx readings and logarithmic gravimetric $PM_{2.5}$ mass concentrations of 0.79. Low-cost optical PM monitors represent an expanding and promising technological domain in household air pollution assessments due to the meaningful spatiotemporal information they can provide; however, site-specific precautions should be made, and duplicate monitors should be deployed.

It is often said that low-cost instruments do not live up to their name when considering the resources needed to quality-check and control the data they record. There is some truth to this concern, especially in highly variable, complex environments rife with confounders. However, technological advances and progress in statistical methods are offering researchers the option to deploy small, efficient sensors at high spatiotemporal resolution to co-measure and integrate potential confounders into subsequent analysis, displacing study resources from up-front capital to data processing and capacity building costs [7]. As this trend continues, the onus is on researchers to disseminate, perpetuate, and improve processing and analysis techniques toward more cost-effective and meaningful research.

**Supplementary Materials:** The following are available online at http://www.mdpi.com/2073-4433/10/7/400/s1.

**Author Contributions:** Conceptualization, M.P.H., K.L.D., and E.R.C.; methodology, E.R.C., M.P.H.; software, E.R.C., A.M. (Anondo Mukherjee), and T.B.; formal analysis, E.R.C.; laboratory, D.P.; visualization, E.R.C.; writing—review and editing, E.R.C., K.L.D., M.P.H., and N.B.; project administration; K.L.D., M.D., D.A., A.M. (Ali Moro), M.P.H., and A.O.

**Funding:** This research was supported by grants from the US National Institutes of Health Clean Cooking Implementation Science Network and the National Science Foundation (SES 1528811). Publication of this article was funded by the University of Colorado Boulder Libraries Open Access Fund.

**Acknowledgments:** A special thanks goes out to all the P3 study participants for their hospitality, patience and respect. Thank you to all the staff and field workers at the Navrongo Health Research Center for their hard work and dedication, without which this study would not have been possible. We are appreciative of Jiayu Li for sharing sensor data with us from Wang et al. Thank you to family, friends, and mentors for support and guidance. This work was reviewed and approved by the Institutional Review Boards at the University of Colorado Boulder and the Navrongo Health Research Center, a part of the Ghana Health Service.

**Conflicts of Interest:** The authors declare no conflicts of interest.

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
