# Peer review of "Kitchen Area Air Quality Measurements in Northern Ghana: Evaluating the Performance of a Low-Cost Particulate Sensor within a Household Energy Study"

_atmosphere, doi:10.3390/atmos10070400_

Round 1

Reviewer 1 Report

This is a very interesting article, which showed the used of HAPEX in monitoring air pollution. The paper enrich the literature about evaluating performance of low cost PM sensors.
The measure of high level of PM in the kitchen is challenging and the authors have applied different strategies to improve the quality of the data and should be commended for that.

However, there is some minor point need to be considered.

1.       The abstract should include more results about the PM2.5­ concentration

2.       Lines 86 – 87 and lines 96 – 108 paragraphs both revealed the objectives of the article. It should be merged and make clear about the objectives of the article.

3.       Line 131 – 143 should be in the result because this reflects a part of your objective (line 86-87). The result session may add 1 more session about the validation of the HAPEX in the lab environment.

4.       Line 171 should be clear about how long the monitor take for each kitchen because, in the result section, the mean of monitoring time was 47 only (line 422) and only 62 houses included

5.       Did all the kitchen were the same as shown in fig 1? All the kitchens were open wide?

6.       Do the authors think that the level of PM exposed by the cook are higher than the point of measurement in Fig. 1 as they are closer to the stove and exposed to air with higher temperature?

7.       The result session using the 48-hour concentration, However, line 422 revealed the mean only 47h. Only the monitoring lower than 41.6h (line 421) were excluded, why? Why not exclude all the monitoring <48 hours?

8.       Line 419 – 428 should be moved to the method session in the data analysis session

9.       For the comparison, the statistical test with p-value should be added (line 433, 436, 543)

10.   Line 454-463 should be careful in presenting the result. Is it appropriate to compare the total 48hour PM2.5 to 24hour average PM2.5 of WHO?

11.   Line 471 – 476 should be moved to method session in the data analysis session

12.   Line 479, 484, 490,543 should add the p-value or 95%CI

13.   Figure in line 556 should be figure 6

14.   Line 581 it should be 95%CI (not 95CI)

15.   Fig.1 and Fig 7: as mentioned above we wonder whether all kitchen are outdoor (or without roof) because the increase in CO and CO2 (presumably due to burning) are not correlated with the temperature peak in Fig. 7?

16.   Fig. 11: can the authors provide some explanation why the slopes are so low?

17.   Line 641 should add the statistical test

18.   Line 629 should be clear on what is environmental factors? Are there only humidity and temperature?

19.   Table 3 should repeated heading after the page break.

20.   Line 656 – 660 should present the 95%CI instead of p-value because the 95%CI give more information. The result also needs to clear about the ref group.

21.   RMSE did not provide for eq4a (line 655)

22.   Line 673: it may an interesting finding to compare about the change/difference of factor coefficients between the eq 4a and 4b

23.   The conclusion did not present clearly the answers to some of the objectives such as the effectiveness of the algorithm.

Reviewer 2 Report

The authors present an evaluation of the performance of a low-cost particulate matter tool applied to kitchen area air quality measurements. The paper is well written, the corrections made to the sensor data are clearly described and the results are thought of interest to a wider audience.

Therefore I recommend that the paper is accepted. Some minor points:

Some parts of the paper are too long (e.g. the section "Modeling gravimetric PM2.5 concentrations and particle coefficients with low-cost sensors") which affects the readability of the paper. Parts could be moved to supplementary section or the main only the best models could be presented.

Figures 9 and 11 are of poor resolution. Please replace. 

Figure 7d: kithen=> kitchen

Page 7: Figure 1: explain (or omit) CCC = 0.78 (95CI: 0,68, 0.85) and CCC - 0.96 (95CI: 0.94, 0.98). Note this should be Figure 6 instead of Figure 1.

 Line 413: Here the abbreviation CCC is used for the first time. Give also the complete term.

Several of the reference are incomplete or unclear (e.g. 19, 49, 50, 51, 54)
